# *Macrovipera lebetinus obtusa* Venom and Its Fractions Affect Human Dermal Microvascular Endothelial and Fibrosarcoma Cells

**DOI:** 10.3390/ijms26083601

**Published:** 2025-04-11

**Authors:** Narine Ghazaryan, Lars Van Werven, Thomas Liepold, Olaf Jahn, Luis A. Pardo, Naira Ayvazyan

**Affiliations:** 1Orbeli Institute of Physiology of NAS RA, Yerevan 0028, Armenia; naringhazaryan@gmail.com; 2Neuroproteomics Group, Department of Molecular Neurobiology, Max Planck Institute for Multidisciplinary Sciences, 37077 Göttingen, Germany; 3Translational Neuroproteomics Group, Department of Psychiatry and Psychotherapy, University Medical Center Göttingen, 37075 Göttingen, Germany; 4Oncophysiology Group, Max Planck Institute for Multidisciplinary Sciences, 37075 Göttingen, Germany; pardo@mpinat.mpg.de

**Keywords:** *Macrovipera lebetinus obtusa*, obtustatin, the human dermal microvascular endothelial cells, fibrosarcoma HT-1080 cells

## Abstract

The venom of *Macrovipera lebetinus obtusa* (MLO) has remarkable properties that are hard to overlook. This venom’s described 38 protein components work synergistically, forming complexes that greatly enhance their combined effectiveness. Previous studies have shown that both crude venom and one of its components, obtustatin, can reduce sarcoma tumors by 50% and 30%, respectively. Obtustatin, a member of the short disintegrin family, inhibits the angiogenic activity of α1β1 integrin, the adhesive receptor of collagen IV. However, the mechanisms of the greater efficacy of the crude venom compared to its isolated components remain unclear. To investigate this, we propose an experimental work to explore the activity of certain low-molecular-weight components of MLO venom. Our in vitro tests on fibrosarcoma (HT-1080) cells using six venom fractions revealed cytotoxic fractions, which, through mass spectrometry, were identified as containing protein classes such as dimeric and short disintegrins, acidic phospholipase A2, and serine proteinases. Notably, these fractions exhibited minimal toxicity to human dermal microvascular endothelial (HDEC) cells, suggesting their potential as a promising candidate for oncotherapy in the future.

## 1. Introduction

Since ancient times, natural products have played a crucial role in human life. Through trial and error, our ancestors identified products from plants and animals that can have healing or other useful properties and passed down this knowledge from generation to generation. Moreover, many contemporary drugs, including such well-known ones as aspirin and penicillin, have also been derived from natural products. In the modern era, distinct areas of chemistry, biology, and medical research are dedicated to identifying valuable natural products, researching their properties and pharmaceutical applications, and their synthesis and modification that could lead to the development of new drugs [1,2].

*Macrovipera* is part of the 20-million-year-old clade of the so-called Old-World Vipers, consisting of the sister genus *Daboia*, *Montivipera*, and *Vipera*. [3]. *Macrovipera* diverged 15 million years ago, with the subsequent evolution of the two species *M. lebetina* (Afghanistan, Iran, India, Pakistan, Turkmenistan, and Uzbekistan) and *M. schweizeri* (the Greek islands of Milos, Kimolos, Polinos, and Siphnos) [4]. *Macrovipera lebetina* further differentiated into the subspecies *M.l. transmediterranea* (Tunisia and Algeria), *M. l. cernovi* (Afghanistan, Iran, India, Pakistan, Turkmenistan, and Uzbekistan), *M. l. lebetina* (Cyprus), *M. l. obtusa* (Armenia, Azerbaijan, Daghestan, Georgia, Kazakhstan, Lebanon, Iran, Iraq, Jordan, Russia, Syria, Turkey, southern Afghanistan, Pakistan (Kashmir), and north India), and *M. l. turanica* (Tajikistan, Turkmenistan, Uzbekistan, northern Afghanistan and Kashmir) [5,6]. Since 2020, the species and its five subspecies have been called *Macrovipera lebetinus* as it was in the original description of Linnaeus (1758) [7].

Due to its wide distribution, *Macrovipera lebetinus obtusa* (MLO) is of the most medicinal value and interest in Southern Europe and the Middle East. The venom consists of 38 protein components that supposedly form complexes with other compounds of the venom cocktail in the organism to achieve high efficiency [8]. Those proteins belong to only a few prominent protein families, including metalloproteinases, disintegrins, serine proteinases, and phospholipases A_2_ isoforms. We previously showed that crude MLO venom and its separate components have valuable properties in combating cancer. More specifically, we have shown that MLO venom suppresses tumors in S-180 sarcoma-bearing mice by 50%, while one of its components-KTS-disintegrin obtustatin by itself is capable of 30% tumor reduction [9,10,11]. Obtustatin, which is one of the shortest disintegrins found so far (41 amino acids long), is unique because it can only be found in MLO venom and because, unlike most of the other monomeric disintegrins, it contains KTS motif in its active site [12,13,14]. The latter makes obtustatin a potent and selective inhibitor of α1β1 integrin, which is a specific receptor for collagen type IV [15,16,17,18]. α1β1 integrin is highly up-regulated by the vascular endothelial growth factor (VEGF) in cultured endothelial cells, resulting in enhanced α1β1-dependent cell spreading on collagen. It offers essential support for VEGF signaling, the migration of endothelial cells, and tumor angiogenesis [19,20,21,22].

However, in the face of the higher efficiency of the crude venom, it is possible that its other components also contribute to the angiostatic (anti-angiogenic) activity in conjunction with obtustatin or through a distinct mechanism of action. Traditionally, the key role in the cytotoxicity of viper venoms is attributed to the predominant enzymatic components of the venom, such as metalloproteases and phospholipase A_2_, while the impact of the smaller components of the venom is rather underestimated. Hence, the proposed study employing Incucyte Systems for Live-Cell Imaging in a real-time mode aims to assess whether low-molecular weight fractions or proteins derived from MLO venom possess independent activity in vitro within HT-1080 (fibrosarcoma) and human dermal microvascular endothelial cells (HMVEC-D). We found six cytotoxic fractions which, according to mass spectrometric protein identification, contained dimeric disintegrins VLO4 and VLO5, dimeric disintegrin Le3 (a disintegrin domain of the PII metalloprotease), short disintegrin obtustatin, PLA_2_, and serine proteinases.

## 2. Results

### 2.1. Fractionation of MLO Venom and Protein Identification

The protein composition of the MLO venom was investigated using fractionation of the crude venom by the reversed phase high-performance liquid chromatography (RP-HPLC), resulting in 38 fractions (Figure 1), which fully corresponded to the results described by Sanz et al. (2008) [8]. The six low-molecular weight fractions, which were supposed to contain disintegrins and PLA_2_ (labeled as Fr1 to Fr6 in the chromatogram), were separated by SDS-PAGE (Figure 2), and bands of interest were subjected to in-gel digestion with trypsin, followed by protein identification using matrix-assisted laser desorption/ionization time-of-flight mass spectrometry (MALDI-TOF-MS).

For database search, we compiled a FASTA file with 1414 vipera protein sequences from UniProtKB (Appendix A). Peptide mass fingerprinting (PMF) and peptide sequencing results are summarized in Table 1 and the original database search algorithm output files are provided in Appendix A.

From the 1414 entries in our UniProtKB-derived database of Vipera protein sequences, only 109 entries comprise *Macrovipera lebetinus* proteins. Out of those, only four proteins belong to the MLO, including short KTS-disintegrin obtustatin [P83469], dimeric disintegrin VLO4 [P0C6A8], and hetero-dimeric disintegrin subunits VLO5A and VLO 5B [P0C6A9 & P0C6B0], which also demonstrate a very high similarity with the functional parts/domains of the PII SVMP lebetase-3 [Q98995]. The MLO venom fractions Fr1 to Fr5 show only one or two major bands in the SDS-PAGE, from which we identified not only all four above-mentioned MLO disintegrins, but also the homologous proteins from sister clade viper genera contained in the current databases. Proteins in these fractions are demonstrating a high degree of similarities with dimeric disintegrins lebein-2α [Q3BK13], ML-G1 [Q1JRG9], lebein-1α [P83253] from *Macrovipera lebetinus transmediterranea*, and dimeric disintegrin VB7(A/B) [P0C6A6, P0C6A7] from *Vipera berus berus* venom. Fraction Fr5 represents the most abundant viper venom acidic phospholipase A_2_ [C3W4R6]. Fraction Fr6 shows high heterogeneity in SDS-PAGE with at least six major bands in the molecular weight region below 60 kDa. Protein identifications from these bands include a mixture of snake venom serine proteinases VLP1 (VSPF5, factor V-activating enzyme) [Q9PT41], VLP/VSP2 [Q9PT40], L-amino acid oxidase (LAO) [P81375], and again acidic phospholipase A_2_ [C3W4R6].

### 2.2. Detection of Cell Death by MLO Crude Venom

First, we used live cell imaging to research the effect of MLO snake venom on the viability of HT-1080 cells in culture. Cell death was detected using Cytotox green, which specifically binds to the DNA of dead cells due to the loss of membrane integrity caused by cell death. The dye is virtually non-fluorescent when not bound to DNA, and live cells do not produce any fluorescence. The fluorescent signal produced is proportional to the cytotoxicity.

As shown in Figure 3, the influence is very potent in the case of 15 μg/mL concentration of MLO snake venom on HT-1080 cells. The cells lose their connection, and morphological change is observed 2 h after adding the venom. The cells merged, and after 10 h, the cells started to die, and all the cells almost died after 14 h. The situation is entirely different in the case of 1.5 μg/mL venom concentration. The cells again lose the junctions, changing morphology. The cell proliferation decreases by 29% and 11% after 10 h and 24 h, respectively, but without cytotoxicity.

The same experimental scheme was applied to detect the effect of the MLO crude venom on the HDEC cells (Figure 4). This time, the lower concentration did not influence the cell proliferation and viability at all, but a ten-times-higher dose of the venom caused mild changes in the cell population. Some small percentage of cells (3.5%) died within the first 10 h, but then the number of dead cells increased until it reached 20% after 24 h.

### 2.3. MLO Venom Fractions Effect on the HMVEC-D Cells

The following experiments were performed with the purified fractions of the MLO venom, the protein content of which was identified by MS analysis. The primary human microvascular endothelial cells lost their connection to each other; a morphological change was observed 2 h later; cells lost the junctions when treated with fractions Fr1 to Fr4 and Fr6, while cells stayed intact in the presence of Fr5, almost like control cells (see Appendix A). The cells merged with each other, but the percentage of dead cells in the first five fractions was even slightly less than in the untreated control cells. Some other picture was observed for fraction Fr6. Cells also seemed intact like all other samples, but after 30 h the number of dead cells increased noticeably, and until 48 h, the number of dead cells was significantly higher compared to other cell cultures. However, the cell death level for all these fractions in the normal endothelial cells, including fraction Fr6, is very low, so we can be sure that no cytotoxicity is detected in these cells (Figure 5).

### 2.4. MLO Venom Fractions Effect on the HT-1080 Cells

We also tested the separated fractions of the MLO venom on the HT-1080 cells. This time, the results were quite drastic for all fractions (Figure 6). The confluence and agglutination of cells were detected almost immediately for all fractions, except for fraction Fr5, whose impact on the morphology and shape of the cells became pronounced only after 12 h. The cells again lose the junctions, changing shape and morphology, and merge with others quite rapidly. Due to the membrane integrity loss, the reagent binds explicitly to dead cells’ DNA, which enhances the fluorescent signal proportionally to the cytotoxicity rate. The impact of fraction Fr4 was significantly less pathological, but compared to the untreated cells, the percentage of cell proliferation was also three times less than in the control (Figure 7A). The cytotoxicity level of all six fractions was remarkably higher, unlike the control group. The highest effect was detected for Fr1, followed by Fr3 and Fr2, and then by Fr4 to Fr6 (Figure 7B).

## 3. Discussion

The medicinal properties of snake venoms have been recognized since the dawn of civilization [23,24,25,26]. Animal venoms are pharmacologically effective at very low doses, and their therapeutic effects arise through mechanisms distinct from those of conventional treatments. The contrast between the toxic effects of purified venom components and the action of the whole cocktail of crude venom is a point of specific interest [27,28]. The synergistic contribution of venom proteins in the course of envenomation may either amplify their effects or facilitate the spread of toxins. Recent research has pointed out the impact of venom components on mammalian cell membranes [29,30]. Much of this research focuses on the role of phospholipase A_2_ (PLA_2_) in the MLO venom’s interaction with membrane lipids. However, purified PLA_2_ is known to have only 1% of the toxic potency found in whole venom. Additionally, low molecular weight components, such as obtustatin, lebein disintegrins, or c-type-lectin-like proteins are also thought to play key roles in envenomation, although the scientific literature on this topic remains inconsistent [31,32].

Viperid venoms may contain well over 100 protein components [1,33]. Most of these proteins can be found also in the venom of other snake families (e.g., *Elapidae*). However, several sets of proteins, both enzymes (serine proteinases, Zn2+-metalloproteinases, L-amino acid oxidase, group II PLA_2_) and non-enzymes (disintegrins, C-type lectins, natriuretic peptides, myotoxins, CRISP toxins, nerve and vascular endothelium growth factors, and Kunitz-type proteinase inhibitors), are uniquely characteristic for the viperid venom with respect to the biochemical and physiological effects. Some proteins found in the venom of MLO are only characteristic of this venom, e.g., the short KTS-disintegrin obtustatin [34,35].

Phospholipase A_2_ (PLA_2_) catalyzes the breakdown of the sn-2 ester bond in phosphoglycerides, releasing free fatty acids and lysophospholipids. These enzymes are found widely across the natural world. In mammals, PLA_2_s are generally non-toxic and play crucial roles in various physiological functions and pathological conditions, such as rheumatoid arthritis, osteoarthritis, asthma, and psoriasis [36]. In contrast, PLA_2_s from snake venoms are among the most potent toxic proteins, often playing a pivotal role in the immobilization and lethality of prey [37]. Within the venom of a single snake, multiple PLA_2_ isoenzymes may exist, each with varying pharmacological effects. Many presynaptic neurotoxins and myotoxins from snake venoms are either PLA_2_s themselves or contain PLA_2_-like subunits in their molecular structure [38].

Contrary to the snake venom PLA_2_s, which are objects of a broad spectrum of investigations for almost a century, disintegrins—specific integrin receptor antagonists—were discovered just recently, in the early 90s of the 20th century [39]. Disintegrins are a diverse group of small, cysteine-rich polypeptides (ranging from 40 to 100 amino acids) that are primarily found in the venoms of vipers and rattlesnakes. These peptides are generated through the proteolytic processing of PII snake venom metalloprotease precursors and selectively inhibit the function of integrin family cell surface adhesion receptors [40]. Disintegrins are categorized based on their peptide length and the number of disulfide bonds present. Functionally, they are divided into three groups according to the specific integrin-binding motif, which governs their selective interactions. The first group consists of disintegrins that bind to RGD-dependent integrins (e.g., αIIβ3, αν-integrins, and α5β1), typically characterized by the presence of the RGD tripeptide in their active site. The second group includes heterodimeric disintegrins that possess the MLD sequence in their binding site and primarily target leukocyte integrins such as α4-integrins and α9β1. The third group comprises disintegrins that selectively inhibit α1β1 integrin, distinguished by the presence of the KTS motif in their active site [41]. The KTS-disintegrin, obtustatin, was isolated from the venom of *Macrovipera lebetinus obtusa* [12]. Structurally, KTS-disintegrins are classified as monomeric short disintegrins, resembling other known short disintegrins like echistatin, lebestatin, or eristostatin. These peptides feature eight cysteine residues, which form four intramolecular disulfide bonds. The 3D structure of obtustatin has recently been determined using NMR data [42].

In our present investigation, the proteomic analysis let us identify RGD and VGD dimeric disintegrins lebein 1-alpha, VLO4, and VLO5 as single components of fractions 1, 2, and 4, respectively. These disintegrins are known as the inhibitors of the human platelet aggregation. They can bind α7/β1, α5/β1, and α4/β1 integrins, respectively, but they can also avidly bind some other receptors of the cell adhesion. However, these types of disintegrin are unable to bind collagen (laminin) receptors, e.g., α1/β1, α2/β1, and α6/β1. In contrast, the short disintegrin of fraction Fr3, known as obtustatin, and present only in the subcaucasian subspecies of the Levantine viper, is a potent and selective inhibitor of the collagen IV receptor α1/β1. It blocks the interaction of the α1β1 integrin with collagens IV and I in vitro with IC50s of 2 nM and 0.5 nM, respectively, and inhibits angiogenesis in vivo [43]. The fraction Fr5 was identified as a nominal acidic phospholipase A_2_, which is also known as a D49 isoform of the snake venom PLA_2_, and this is one of the predominant components of viper venoms, especially of the so-called Old-World vipers (*Macrovipera*, *Montivipera*, *Vipera*). In the venom of the MLO, the PLA_2_ content is about 14% of the dry venom weight, the second most abundant component after Zn-dependent metalloproteases. MLO PLA_2_s, the products of the six different gens in the MLO venom, share this second place with serine proteinases and C-type lectin-like peptides [8]. The same acidic PLA_2_ is also present in fraction Fr6, which is a mixture of a few different venom components, also including the above-mentioned serine proteinases (VSP1, VSP2, and Factor V-activator), and the LAAO.

It is intriguing that neither PLA_2_ (Fr5) nor its mixture with SVSP demonstrated such toxic effects on the cells as disintegrins did. Moreover, the latter showed remarkable and pronounced cytotoxic action in the tumor cells but not in the culture of the normal endothelial vascular cells. This finding confirms the investigators’ interest in these novel compounds in the aspect of their future anticancer drug potential. Several studies show their potential as anti-metastatic agents [10,32]. For example, the venom of the Levantine viper has been shown to induce apoptosis in melanoma cells [11] and inhibit human colon adenocarcinoma [44]. Additionally, the homodimeric CC5 and heterodimeric CC8 proteins from the venom of the desert horned viper, *Cerastes cerastes*, are known to inhibit tumor angiogenesis [45]. Several monomeric disintegrins from the *Crotalus*, such as r-mojastin 1, r-viridistatin 2, and tzabcanin, have also demonstrated anti-cancer properties [46,47]. Moreover, some recent studies suggest that disintegrins hold potential not only as powerful antiangiogenic agents for cancer therapy but also for treating ocular diseases (such as diabetic retinopathy) and as proangiogenic agents to promote tissue regeneration and wound healing [22,32]. Unfortunately, from the literature reports, it seems that the interest in snake venom disintegrins peaked at the beginning of the 21st century and noticeably decreased within the last decade.

Nevertheless, based on our and other toxinological investigations, we firmly believe that disintegrins and obtustatin, in particular, may serve as a promising lead compound for developing innovative anti-angiogenic therapies for aggressive cancers. Investigating the interactions of venom components with cells and tissues in a potent, specific, selective manner is of considerable theoretical interest for understanding the molecular evolution mechanisms, designing novel therapeutic agents, and developing effective antivenom production strategies.

## 4. Materials and Methods

### 4.1. RP-HPLC for the Separation of MLO Venom Components

MLO venom was purchased from Latoxan. RP-HPLC separations were, in principle, performed as described in Sanz et al., 2008 [8]. Briefly, 1 mg of crude, lyophilized venom was dissolved in 100 μL of 0.05% trifluoroacetic acid (TFA) and 5% acetonitrile, and insoluble material was removed by centrifugation in an Eppendorf centrifuge at 13,000× *g* for 10 min at room temperature. Soluble proteins in the supernatant were separated using a Beckmann HPLC Gold System (Pump: 126 Solvent Module, Detector: 168 PDA 200–600 nm, Software: 32 Karat 5.0), equipped with a Vydac 218TP54 C18 column of the dimensions 4.6 × 250 mm. The following gradient was performed with a flow rate of 1 mL/min: 10 min at 5% B, 20 min 5% B to 15% B, 120 min 15% B to 45% B, 20 min 45% B to 70% B, 10 min 70% B to 95% B, 5 min at 95% B. Mobile phases composed of 0.1% TFA and 80% acetonitrile/0.1% TFA were used as solvent A and solvent B, respectively.

### 4.2. Gel Electrophoresis

MLO venom proteins were separated under reducing conditions using precast NuPAGE 4–12% Bis-Tris gradient gels (1 mm, 10 wells, Thermo Fisher Scientific, Waltham, MA, USA) in MES running buffer. SeeBlue-Plus-2 (Thermo Fisher Scientific, Waltham, MA, USA) served as the molecular weight marker. The gels were fixed for 60 min in a solution of 40% (*v*/*v*) ethanol and 10% (*v*/*v*) acetic acid, followed by three 10-min rinses in water. Subsequently, the gels were stained overnight with colloidal Coomassie staining solution composed of 0.08% (*w*/*v*) Coomassie Brilliant Blue G-250, 1.6% (*w*/*v*) ortho-phosphoric acid, 8% (*w*/*v*) ammonium sulfate, and 10% (*v*/*v*) methanol. Bands of interest were manually excised from the gel.

### 4.3. Mass Spectrometric Protein Identification

Proteins in gel bands were subjected to in-gel digestion with trypsin followed by an analysis of the proteolytic peptides with MALDI-TOF-MS as described in detail earlier [48,49]. A Bruker UltrafleXtreme MALDI-TOF-TOF mass spectrometer operated under flexControl 3.4 was employed to obtain both peptide mass fingerprint (PMF) and fragment ion spectra, enabling reliable protein identifications based on peptide mass and sequence data. Database searches were conducted using the MASCOT Software version 2.3.02 (Matrix Science, London, UK) against the UniProtKB-derived vipera sequence database (1414 entries, Appendix A). Carboxamidomethylation of cysteine residues was set as a fixed modification, while methionine oxidation was considered variable. Trypsin was selected as the protease, with allowance for one missed cleavage. Mass tolerance parameters were set to 100 ppm for PMF searches and 100 ppm for precursor ions and 0.7 Da-for fragment ions in MS/MS ion searches. A protein was deemed identified if at least one peptide sequence match above the identity threshold was observed, alongside a minimum of 20% sequence coverage in the PMF.

### 4.4. IncuCyte Cytotox Green Detection of Cell Death

The IncuCyte Cytotox green reagent was used according to the manufacturer’s instructions to quantify dead cells after 48 h of treatment using six fractions isolated from the MLO venom. The HT-1080 (fibrosarcoma) and human dermal microvascular endothelial (HMVEC-D) cells were used, and image fields were acquired with Incucyte SX1Systems for Live-Cell Imaging (Sartorius, Göttingen, Germany). The experiments were performed in triplicates. The fractions were diluted in the medium where the cells were grown. The HT-1080 (Passage 5–9) cells were grown in DMEM medium supplemented with 5 g/L Glucose, L- Glutamine, sodium pyruvate, and 10% fetal bovine serum (FBS) under a 5% CO_2_-humidified atmosphere at 37 °C. Primary adult HMVEC-D cells were purchased from ATCC (Manassas, VA, USA) and cultured in complete endothelial cell basal 131 medium supplemented with microvascular growth supplement (MVGS) containing 15% FBS (all from Invitrogen, Carlsbad, CA, USA). Early passage (3–6) HMVEC-D cells were used in all experiments. The recombinant human vascular endothelial growth factor (VEGF) and collagen IV were obtained from Sigma (St. Louis, MO, USA); the fibroblast growth factor (FGF2) was purchased from Immuno Tools (Friesoythe, Germany).

### 4.5. Statistical Analysis

For quantitative analysis of proliferation and cytotoxicity rate, results are reported as means ± SEM. The significance of differences between the means was assessed one-way ANOVA with Dunnett’s multiple comparison test, when each of the various experimental groups was compared with the control group. A value of * *p* < 0.05 indicated significance. All statistical analyses were performed with Prism version 8.0.1 for Windows, GraphPad Software, La Jolla, CA, USA.

## Figures and Tables

**Figure 1 ijms-26-03601-f001:**
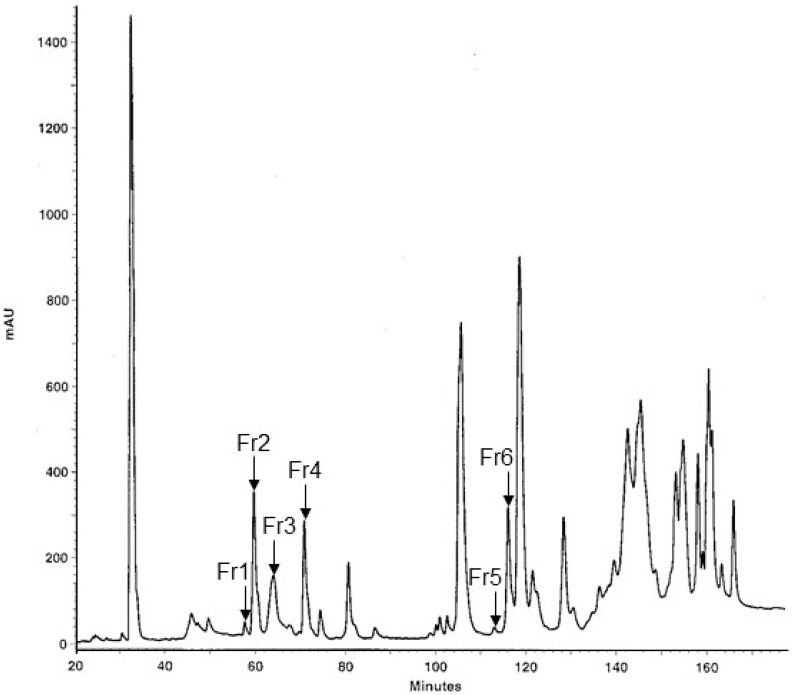
RP-HPLC separation of the proteins from *Macrovipera lebetinus obtusa* snake venom. The chromatogram shows absorbance at 220 nm. The fractions to be subjected to cell assays and protein identification, respectively, were collected manually and are labeled Fr1 to Fr6.

**Figure 2 ijms-26-03601-f002:**
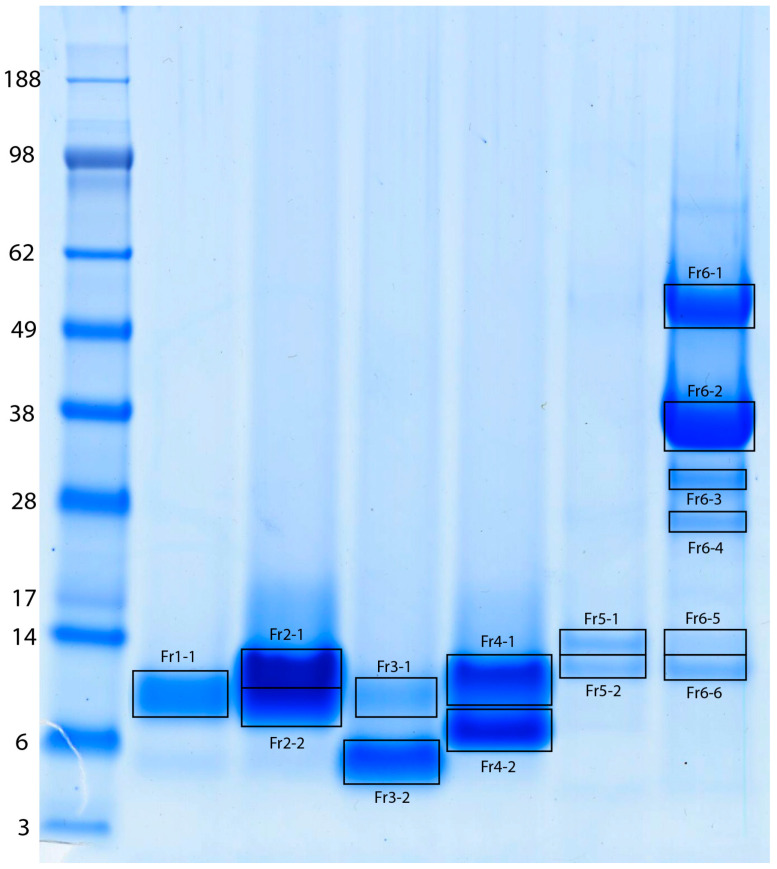
SDS-PAGE separation of proteins contained in the six active MLO venom fractions isolated by HPLC as shown in Figure 1. Bands of interest were excised as labeled in the gel and their protein content was identified by in-gel protein digestion with trypsin followed by MALDI-TOF-MS. The leftmost lane shows a molecular weight marker with protein sizes assigned in kDa.

**Figure 3 ijms-26-03601-f003:**
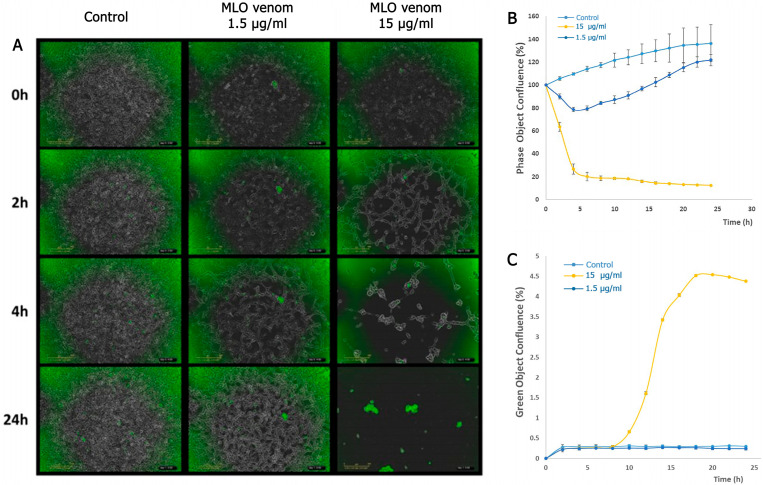
IncuCyte ZOOM^®^ analysis for HT-1080 (fibrosarcoma) cells behavior induced by MLO snake venom (1.5 μg/mL and 15 μg/mL) after 24 h treatment. (**A**) Cytotox green (0.25 µM) was added after a 24 h treatment with MLO snake venom. The representative field images were acquired with IncuCyte ZOOM^®^. Scale bar, 300 µm (**B**,**C**). The rate of the proliferation and cytotoxicity of the HT-1080 cells after treatment with two different concentrations with MLO snake venom for 24 h (15 μg/mL and 1.5 μg/mL) (experimental repeats–3, mean ± SEM).

**Figure 4 ijms-26-03601-f004:**
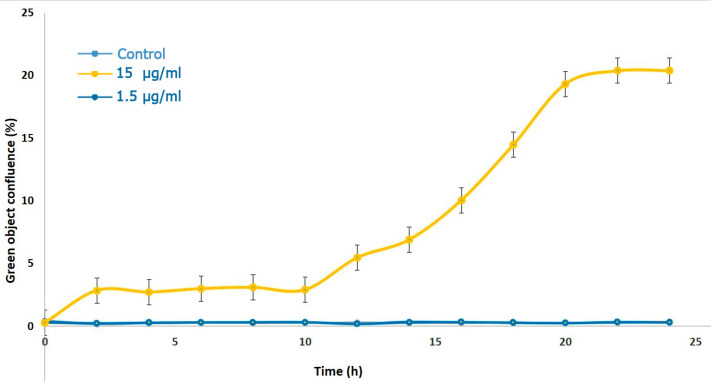
The rate of the cytotoxicity of the HDEC cells after treatment with two different concentrations of MLO snake venom for 24 h (15 μg/mL and 1.5 μg/mL) (experimental repeats–3, mean ± SEM).

**Figure 5 ijms-26-03601-f005:**
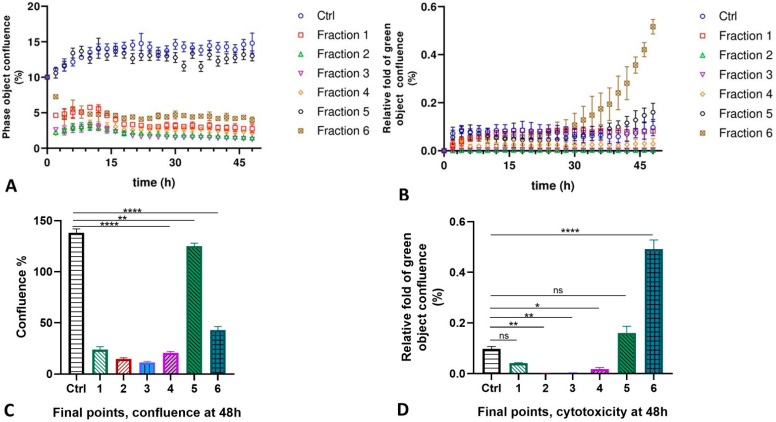
The rate of the proliferation (panels **A**,**C**) and cytotoxicity (panels **B**,**D**) of the HMVEC-D cells after treatment with six fractions isolated from the MLO snake venom for 48 h (experimental repeats–3, mean ± SEM; one-way ANOVA with Dunnett’s multiple comparison test, *p* * < 0.05; *p* ** < 0.01; *p* **** < 0.0001; ns—not significant).

**Figure 6 ijms-26-03601-f006:**
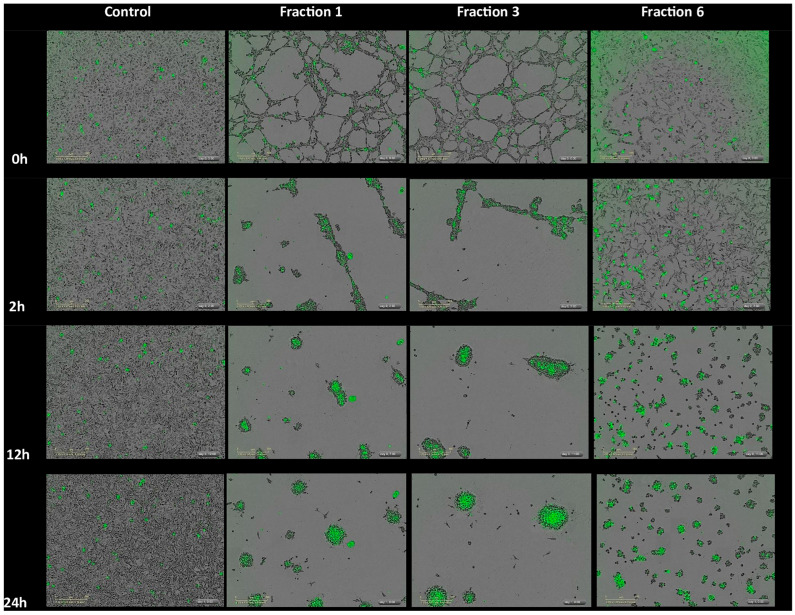
IncuCyte ZOOM^®^ analysis for HT-1080 cells behavior induced by some MLO active fractions after 24 h treatment. Cytotox green (0.25 µM) was added after a 24 h treatment with fractions. The representative field images were acquired with IncuCyte ZOOM^®^. Scale bar, 300 µm. The IncuCyte^®^ Cytotox green reagent specifically binds to DNA of dead cells due to the loss of membrane integrity. Hence, the fluorescent signal produced is proportional to the cytotoxicity.

**Figure 7 ijms-26-03601-f007:**
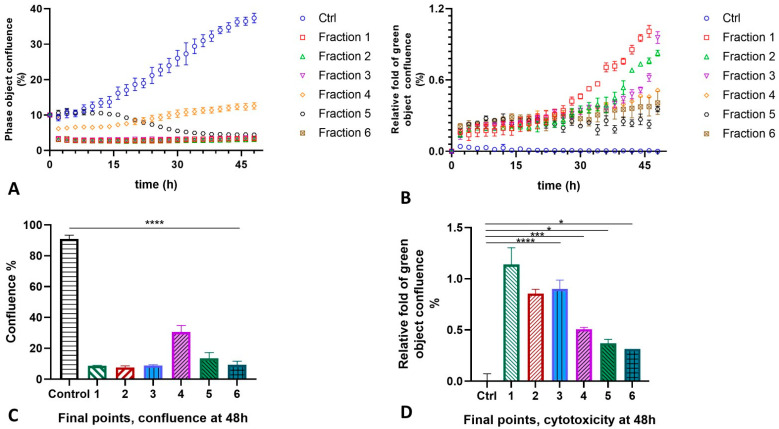
The rate of the proliferation (panels **A**,**C**) and cytotoxicity (panels **B**,**D**) of the HT-1080 cells after treatment with 6 fractions isolated from the MLO snake venom for 48 h (experimental repeats – 3, mean ± SEM; one-way ANOVA with Dunnett’s multiple comparison test, *p* * < 0.05; *p* *** < 0.001; *p* **** < 0.0001; ns—not significant).

**Table 1 ijms-26-03601-t001:** Protein identification data.

Gel Band	Protein Name	UniProtKB Accession	UniProtKB ID	Species	MW [Da]	pI	PMF Score ^(a)^	PMF % s.c.	Peptide Sequenced	MS/MS Ion Score ^(b)^
Fr1-1	Disintegrin lebein-1-alpha	P83253	DID1A	MACLB	12,718	7.42	46	39	K.FLNAGTICNR.A R.RGEHCVSGPCCR.N R.GDDMNDYCTGISSDCPR.N	71 62 141
	VGD-containing dimeric disintegrin subunit ML-G1 (Fragment)	Q1JRG9	Q1JRG9	MACLN	7698	8.05	36	56	R.RGEHCVSGPCCR.N R.AVGDDMDDYCTGISSDCPR.N	62 146
Fr2-1	Disintegrin lebein-1-alpha	P83253	DID1A	MACLB	12,718	7.42	57	39	K.FLNAGTICNR.A R.RGEHCVSGPCCR.N R.GDDMNDYCTGISSDCPR.N	70 60 144
	VGD-containing dimeric disintegrin subunit ML-G1 (Fragment)	Q1JRG9	Q1JRG9	MACLN	7698	8.05	61	90	R.RGEHCVSGPCCR.N R.AVGDDMDDYCTGISSDCPR.N	60 151
Fr2-2	Disintegrin lebein-1-alpha	P83253	DID1A	MACLB	12,718	7.42	67	39	K.FLNAGTICNR.A R.GEHCVSGPCCR.N R.RGEHCVSGPCCR.N R.GDDMNDYCTGISSDCPR.N	71 34 60 142
	Disintegrin VB7A	P0C6A6	DID7A	VIPBB	7574	8.10	76	68	R.GEHCVSGPCCR.N R.RGEHCVSGPCCR.N -.NSGNPCCDPVTCKPR.R R.GDDMNDYCTGISSDCPR.N	34 60 80 142
	Disintegrin VLO4	P0C6A8	DID4	MACLO	7673	8.60	71	84	R.GEHCVSGPCCR.N R.RGEHCVSGPCCR.N -.NSGNPCCDPVTCKPR.R -.MNSGNPCCDPVTCKPR.R	34 60 80 77
	VGD-containing dimeric disintegrin subunit ML-G1 (Fragment)	Q1JRG9	Q1JRG9	MACLN	7698	8.05	51	70	R.GEHCVSGPCCR.N R.RGEHCVSGPCCR.N R.AVGDDMDDYCTGISSDCPR.N	34 60 110
Fr3-1	Disintegrin lebein-1-alpha	P83253	DID1A	MACLB	12,718	7.42	49	56	K.FLNAGTICNR.A R.GEHCVSGPCCR.N R.RGEHCVSGPCCR.N R.GDDMNDYCTGISSDCPR.N	74 22 48 109
	Disintegrin VB7A	P0C6A6	DID7A	VIPBB	7574	8.10	57	68	R.GEHCVSGPCCR.N R.RGEHCVSGPCCR.N -.NSGNPCCDPVTCKPR.R R.GDDMNDYCTGISSDCPR.N	22 48 55 109
Fr3-2	Disintegrin obtustatin	P83469	DIS	MACLO	4854	8.72	57	70	-.CTTGPCCR.Q K.LKPAGTTCWK.T K.TSLTSHYCTGK.S K.LKPAGTTCWKTSLTSHYCTGK.S	15 68 60 24
Fr4-1	Disintegrin VLO5A	P0C6A9	VM25A	MACLO	7570	8.11	80	87	R.NCKFLR.A R.RGEHCVSGK.C -.NSGNPCCDPVTCQPR.R R.AVGDDMDDYCTGISSDCPR.N K.RAVGDDMDDYCTGISSDCPR.N	24 31 36 151 58
	Disintegrin VB7B	P0C6A7	VM27B	VIPBB	7563	7.71	23	57	-.ELLQNSGNPCCDPVTCKPR.E	75
Fr4-2	Disintegrin VLO5B	P0C6B0	DID5B	MACLO	8235	6.52	88	81	K.FLNPGTICK.R K.FLNPGTICKR.T R.GEHCVSGPCCR.N M.NSANPCCDPITCKPR.R -.MNSANPCCDPITCKPR.R R.TMLDGLNDYCTGVTSDCPR.N	53 30 25 54 83 155
	Disintegrin lebein-2-alpha	Q3BK13	DID2A	MACLB	14,638	7.45	43	32	K.FLNPGTICK.R K.FLNPGTICKR.T K.GEHCVSGPCCR.N R.KGEHCVSGPCCR.N	53 30 25 80
Fr5-1/2	Acidic phospholipase A_2_ 1	C3W4R6	PA2A1	MACLB	16,337	4.65	71	63	K.YMLYSLFDCK.E K.YMLYSLFDCKEESEK.C R.CCFVHDCCYGSVNGCDPK.L K.LSTYSYSFQNGDIVCGDDDPCLR.A	52 24 43 120
Fr6-1	Venom serine proteinase-like protein 2	Q9PT40	VSP2	MACLB	29,559	9.07	40	29	R.FYCAGTLINQEWVLTAAR.C	48
Fr6-2	Venom serine proteinase-like protein 2	Q9PT40	VSP2	MACLB	29,559	9.07	61	38	R.WDKDIMLIR.L K.NVPNEDQQIR.V R.TLCAGILQGGIDSCK.V R.FYCAGTLINQEWVLTAAR.C K.VTYPDVPHCANINMFDYSVCR.K	81 43 110 174 132
	Snake venom serine protease VaSP1 (Fragments)	P0DPS3	VASP1	VIPAA	22,635	8.65	28	32	R.WDKDIMLIR.L R.TLCAGILQGGIDSCK.G R.FYCAGTLINQEWVLTAAR.C -.VIGGDECNINEHPFLVALHTAR.X	81 110 174 176
Fr6-3	Factor V activator	Q9PT41	VSPF5	MACLB	29,,261	8.81	68	46	R.NEDEQIR.V K.FFCLNTK.F R.TLCAGILQGGK.D K.NSAHIAPISLPSSPSSPR.S K.ISTTEETYPDVPHCAK.I K.HAWCEALYPWVPADSR.T K.DTCEGDSGGPLICNGQIQGIVSGGSDPCGQR.L	14 30 47 130 70 102 164
	Snake venom serine protease VaSP1 (Fragments)	P0DPS3	VASP1	VIPAA	22,635	8.65	29	32	R.FYCAGTLINQEWVLTAAR.C -.VIGGDECNINEHPFLVALHTAR.X	76 129
	Venom serine proteinase-like protein 2	Q9PT40	VSP2	MACLB	29,559	9.07	66	45	R.FYCAGTLINQEWVLTAAR.C K.VTYPDVPHCANINMFDYSVCR.K	76 43
Fr6-4	Venom serine proteinase-like protein 2	Q9PT40	VSP2	MACLB	29,559	9.07	56	32	K.FFCLSSK.T R.WDKDIMLIR.L R.IMGWGTITTTK.V K.NVPNEDQQIR.V R.FYCAGTLINQEWVLTAAR.C K.VTYPDVPHCANINMFDYSVCR.K	26 81 41 29 115 97
	Snake venom serine protease VaSP1 (Fragments)	P0DPS3	VASP1	VIPAA	22,635	8.65	22	24	R.WDKDIMLIR.L R.FYCAGTLINQEWVLTAAR.C -.VIGGDECNINEHPFLVALHTAR.X	81 115 139
	Factor V activator	Q9PT41	VSPF5	MACLB	29,261	8.81	20	22	K.NSAHIAPISLPSSPSSPR.S	108
Fr6-5	L-amino-acid oxidase (Fragments)	P81375	OXLA	MACLB	12,541	4.52	23	31	-.ADDKNPLEECFR.E	47
	L-amino-acid oxidase	G8XQX1	OXLA	DABRR	57,251	8.55	41	21	K.EGWYANLGPMR.V	40
Fr6-6	Acidic phospholipase A_2_ 1	C3W4R6	PA2A1	MACLB	16,337	4.65	104	63	K.YMLYSLFDCK.E R.VAAICFGENMNTYDK.K K.YMLYSLFDCKEESEK.C R.CCFVHDCCYGSVNGCDPK.L K.LSTYSYSFQNGDIVCGDDDPCLR.A	56 87 44 31 98
MACLB—*Macrovipera lebetinus*, MACLN—*Macrovipera lebetinus transmediterranea*, VIPBB—*Vipera berus berus*, MACLO—*Macrovipera lebetinus obtusa*, VIPAA—*Vipera ammodytes ammodytes*, DABRR—*Daboia russelii*										

Proteins were accepted as identified with at least one peptide sequence match above identity threshold in addition to 20% sequence coverage in PMF. (a) MASCOT protein score obtained for the PMF. The significance threshold was 44. (b) MASCOT MS/MS ion score obtained for the individual peptide sequenced. The significance threshold for identity was 10–14, depending on how many peptides fell within the mass tolerance window around the precursor mass. Only the top-ranking peptide matches matching a query for the first time (‘bold red hits’) are reported.

## Data Availability

The original contributions presented in this study are included in the article/Appendix A. Further inquiries can be directed to the corresponding author(s).

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
