# Peer review of "Macrovipera lebetinus obtusa Venom and Its Fractions Affect Human Dermal Microvascular Endothelial and Fibrosarcoma Cells"

_ijms, 2025, doi:10.3390/ijms26083601_

Round 1

Reviewer 1 Report

Comments and Suggestions for Authors
  • In the title the scientific name must be in cursive letter, and the other hand, the authors must use the actual taxonomy Macrovipera lebetina obtusa is a synonymous of the Macrovipera lebetinus.

  • In line 14 use the ideal taxonomy.

  • In line 22, change experimental study by experimental work.

  • In line 30, use the ideal taxonomy.

  • In line 40, change studying by researching.

  • On introduction, in line 47, use the ideal taxonomy and change in all manuscript Macrovipera lebetina obtusa by Macrovipera lebetinus.

  • On introduction, the authors must include a paragraph over the actual taxonomy of the Macrovipera lebetinus and explain why Macrovipera lebetina obtusa is a synonymous of the Macrovipera lebetinus or to justify why to use Macrovipera lebetina obtusa.

  • In line 130, change study by research or work.

  • On figure 5, in the panels C and D, the authors must explain why not using the ANOVA test or its nonparametric equivalent the Kruskal Wallis test.

  • On figure 6, in the legend, the authors must modify it and explain it better.

  • On figure 7, in the panels C and D, the authors must explain why not using the ANOVA test or its nonparametric equivalent the Kruskal Wallis test.

  • In line 203, change studies by works or researches.

  • In line 209, change intoxication by envenoming.

  • In line 209, use viperid venoms, Crotalidae now is Crotalinae that is a subfamily of the Viperidae.

  • The authors must strengthen the discussion, focusing on the antitumor activity of the molecules they identified in this work, as well as indicating which are the most promising molecules.

  • On statistical analysis section, the authors must say if the hypothesis contrast was bilateral or unilateral and why use the t-test and no the its nonparametric equivalent U Mann Whitney test, as well as, to explain why no use an variance analysis when groups are compared.

Author Response

Honorable sir,
Thank you very much for the detailed analysis of our work: indeed, all reviewer's remarks are very relevant and constructive. Now, in this annotated letter, let us describe point-by-point any corrections we've made according to the issues raised in your comments, addressing each point individually.

Comments 1: In the title the scientific name must be in cursive letter, and the other hand, the authors must use the actual taxonomy Macrovipera lebetina obtusa is a synonymous of the Macrovipera lebetinus.

Response 1: The snake's scientific name in the title changed to the cursive letters. Indeed, the taxonomy very recently (2020) been changed from Macrovipera lebetina to M. lebetinus, but actually, Macrovipera lebetina obtusa is not synonymous with just Macrovipera lebetinus. Macrovipera lebetinus viper species are divided into five subspecies with very different venom compositions (in some cases, like for M.l.transmediterranea vs M.l.obtusa, the similarity coefficient is only 4%). So it is absolutely crucial to mention the exact subspecies of the snake, mainly because among the fractionated components is the short disintegrin obtustatin presented in the venom of only this particular subspecies.

Comments 2: In line 14 use the ideal taxonomy.

Response 2: Changed to the Macrovipera lebetinus obtusa.

Comments 3: In line 22, change experimental study by experimental work.

Response 3: Done.

Comments 4: In line 30, use the ideal taxonomy.

Response 4: Changed to the Macrovipera lebetinus obtusa.

Comments 5: In line 40, change studying by researching.

Response 5: Done.

Comments 6: On introduction, in line 47, use the ideal taxonomy and change in all manuscript Macrovipera lebetina obtusa by Macrovipera lebetinus.

Response 6: Changed to the Macrovipera lebetinus obtusa (see above the description of this choice).

Comments 7: On introduction, the authors must include a paragraph over the actual taxonomy of the Macrovipera lebetinus and explain why Macrovipera lebetina obtusa is a synonymous of the Macrovipera lebetinus or to justify why to use Macrovipera lebetina obtusa.

Response 7: We added a paragraph that describes the taxonomy, evolution, and distribution of all five subspecies of Macrovipera lebetinus and which one and why is the exact object of interest for this investigation. 

Comments 8: In line 130, change study by research or work.

Response 3: Done.

Comments 9: On figure 5, in the panels C and D, the authors must explain why not using the ANOVA test or its nonparametric equivalent the Kruskal Wallis test.

Response 9: We intended not to directly compare different fractions but to compare each to the control. Therefore, we found it applicable to use several t-tests comparing the control to the others rather than ANOVA. However, considering that this point was mentioned by reviewers and the academic editor as confusing, we replaced the data from pictures with those statistically analyzed with one-way ANOVA.

Comments 10: On figure 6, in the legend, the authors must modify it and explain it better.

Response 10: We made the description more detailed.

Comments 11: On figure 7, in the panels C and D, the authors must explain why not using the ANOVA test or its nonparametric equivalent the Kruskal Wallis test.

Response 11: We intended not to directly compare different fractions but to compare each to the control. Therefore, we found it applicable to use several t-tests comparing the control to the others rather than ANOVA. However, considering that this point was mentioned by reviewers and the academic editor as confusing, we replaced the data from pictures with those statistically analyzed with one-way ANOVA. 

Comments 12: In line 203, change studies by works or researches.

Response 12: Done.

Comments 13: In line 209, change intoxication by envenoming.

Response 13: Done.

Comments 14: In line 209, use viperid venoms, Crotalidae now is Crotalinae that is a subfamily of the Viperidae.

Response 14: Done.

Comments 15: The authors must strengthen the discussion, focusing on the antitumor activity of the molecules they identified in this work, as well as indicating which are the most promising molecules.

Response 15: We generally rewrote the Discussion to address this remark and better explain the translational perspectives of the disintegrins from viperid venoms, particularly MLO venom. 

Comments 16: On statistical analysis section, the authors must say if the hypothesis contrast was bilateral or unilateral and why use the t-test and no the its nonparametric equivalent U Mann Whitney test, as well as, to explain why no use an variance analysis when groups are compared.

Response 16: Well, our hypothesis is rather unilateral, so we found that t-test is appropriate, and these data meet certain assumptions, particularly normality and homogeneity of variance. Moreover, for each fraction, we compared its level of cytotoxicity and confluence action with only the control group, so the t-test is quite appropriate for this case, especially because we tried to keep figures 5 and 7 as easily understandable as possible. Yet, we have done the one-way ANOVA analysis at the request of the academic editor, and the results are almost identical to those previously presented in the manuscript.

Reviewer 2 Report

Comments and Suggestions for Authors

This is a well-written article on the effects of MLO venom and its fractions on two different cell types. The authors have characterised the fractions isolated from this venom and studied their impacts. Overall, it is a useful article to report some novel findings, however, I suggest authors address the following comments. 

  1. There are some typographical errors in the manuscript, so I suggest authors to address them all by carefully proofreading the manuscript.
  2. In PLA2, 2 should be in subscript consistently across the manuscript.
  3. The title should read as 'Macrovipera lebetina obtusa venom and its fractions affect human dermal microvascular endothelial and gibrosarcoma cells'.
  4. Please check the spelling for 'domain'.
  5. it should read as 'low molecular weight fractions or proteins.
  6. RP-HPLC should be used consistently.
  7. In the results section, please describe only the results here. No need to provide any methods or introduction here. 
  8. Were the concentrations of venom used ug/mL or ug/uL? If it was the latter, then please change this to mL as a standard term. 
  9. Please highlight the statistical significance and experimental repeats in all figure legends and figures as necessary. 
  10. Please see if the data in figures 5 and 7 can be presented differently instead of showing confluence and relative fold change in fluorescence. The key findings are confusing here. So please present the data differently to demonstrate the cytotoxic effects more clearly if possible. 
  11. Some mechanistic details will be helpful in demonstrating the impacts of specific fractions on fibrosarcoma cells. 

Author Response

Honorable sir,
Thank you very much for the positive assessment and detailed analysis of our work: all reviewer's remarks are very relevant and constructive. Now, in this annotated letter, let us describe point-by-point any corrections we've made according to the issues raised in your comments, addressing each point individually.

Comments 1: There are some typographical errors in the manuscript, so I suggest authors to address them all by carefully proofreading the manuscript.

Response 1: We corrected all such errors, which are primarily typos.

Comments 2: In PLA2, 2 should be in subscript consistently across the manuscript.

Response 2: Done.

Comments 3: The title should read as 'Macrovipera lebetina obtusa venom and its fractions affect human dermal microvascular endothelial and gibrosarcoma cells'.

Response 3: Agree, done (fibrosarcoma, of course).

Comments 4: Please check the spelling for 'domain'.

Response 4: Done.

Comments 5: it should read as 'low molecular weight fractions or proteins.

Response5: Done.

Comments 6: RP-HPLC should be used consistently.

Response 6: Done.

Comments 7: In the results section, please describe only the results here. No need to provide any methods or introduction here. 

Response 7: We have done some paraphrase and excluded such sentences from the Results to meet this remark and avoid the repeating.

Comments 8: Were the concentrations of venom used ug/mL or ug/uL? If it was the latter, then please change this to mL as a standard term. 

Response 8: Done.

Comments 9: Please highlight the statistical significance and experimental repeats in all figure legends and figures as necessary. 

Response 9: We added the requested information to the Figures where it was lacking.

Comments 10: Please see if the data in figures 5 and 7 can be presented differently instead of showing confluence and relative fold change in fluorescence. The key findings are confusing here. So please present the data differently to demonstrate the cytotoxic effects more clearly if possible. 

Response 10: For this data obtained from the same real-time videos, we found it crucial to present the analyzed data in comparison to each other for two cell cultures. Therefore, we strongly believe they should be parts of the same Figure. Yet, if the respective Reviewer finds it confusing, we added more detailed explanations of the presented data both in the Figure’s legends and in the text.

Comments 11: Some mechanistic details will be helpful in demonstrating the impacts of specific fractions on fibrosarcoma cells. 

Response 11: We made the description more detailed.

Round 2

Reviewer 2 Report

Comments and Suggestions for Authors

The authors have addressed all my concerns adequately. 

Author Response

Comments 1: The authors do not indicate the post hoc test used with ANOVA to compare conditions to the baseline conditions. They must use Dunnett's test or Bonferroni's test, and I prefer Dunnett's test. Please modify the text accordingly.

Response 1: The ANOVA with Dunnett's multiple comparison test was done, and the text and Figures' legends have been changed accordingly (see the attached file for calculations).
